

# Crowdsourcing visual perception experiments: a case of contrast threshold

Kyoshiro Sasaki[1,2,3] and Yuki Yamada[2]

[1] Faculty of Science and Engineering, Waseda University, Tokyo, Japan
[2] Faculty of Arts and Science, Kyushu University, Fukuoka, Japan
[3] Japan Society for the Promotion of Science, Tokyo, Japan

## ABSTRACT

Crowdsourcing has commonly been used for psychological research but not for studies on sensory perception. A reason is that in online experiments, one cannot ensure that the rigorous settings required for the experimental environment are replicated. The present study examined the suitability of online experiments on basic visual perception, particularly the contrast threshold. We conducted similar visual experiments in the laboratory and online, employing three experimental conditions. The first was a laboratory experiment, where a small sample of participants ($n = 24$; laboratory condition) completed a task with 10 iterations. The other two conditions were online experiments: participants were either presented with a task without repetition of trials ($n = 285$; online non-repetition condition) or one with 10 iterations ($n = 166$; online repetition condition). The results showed significant equivalence in the contrast thresholds between the laboratory and online repetition conditions, although a substantial amount of data needed to be excluded from the analyses in the latter condition. The contrast threshold was significantly higher in the online non-repetition condition compared with the laboratory and online repetition conditions. To make crowdsourcing more suitable for investigating the contrast threshold, ways to reduce data wastage need to be formulated.

## INTRODUCTION

Over the last decade, experiments in psychological research have gone beyond the laboratory. The increasing diversity of research methods and technological advances has increased opportunities for researchers to use resources outside the laboratory. For example, researchers are using outsourcing services to recruit experimental participants and, often, even commissioning research firms to conduct their surveys and experiments. In addition, based on outstanding technological advances in the digital environment and mobile information devices, "crowdsourcing," which is the practice of asking many unspecified people to various kinds of tasks via the internet, has become a powerful tool for psychological research (for a review, see *Stewart, Chandler & Paolacci, 2017*).

Crowdsourcing can be used for data collection and in asking large numbers of people to participate in surveys or experiments via the internet. Service providers (e.g., Amazon and

Corresponding author
Yuki Yamada,
yy@artsci.kyushu-u.ac.jp

Yahoo!) manage an experimenter's task and act as a payment agency. The use of crowdsourcing has a number of advantages. The first is its very low cost (*Stewart, Chandler & Paolacci, 2017*); for example, participants receive less than one USD for responding to a simple questionnaire or engaging in an easy cognitive task. Second, large (more than 1,000 people) and diverse (in age, sex, and culture) samples can easily be employed. The ease in collecting large amounts of diverse data is beneficial not only from the perspective of random sampling but also for planning experiments and estimating the effect size prior to conducting the experiment (*Chrabaszcz, Tidwell & Dougherty, 2017*). Third, it enables researchers to use their time efficiently. With experiments running all hours of the day and night, data from 1,000 people can be obtained within a day or two, depending on how many active users are registered with the service.

Various kinds of online experiments and tasks have been conducted with crowdsourcing. For example, many experimental studies have reported findings based on self-report questionnaires (*Crangle & Kart, 2015*; *Garcia et al., 2016*; *Gottlieb & Lombrozo, 2018*; *Hurling et al., 2017*; *Sasaki, Ihaya & Yamada, 2017*) and crowdsourced tasks: visual search (*de Leeuw & Motz, 2016*), reaction time (*Nosek, Banaji & Greenwald, 2002*; *Sasaki, Ihaya & Yamada, 2017*; *Schubert et al., 2013*), keystroke (*Pinet et al., 2017*), Stroop (*Barnhoorn et al., 2015*; *Crump, McDonnell & Gureckis, 2013*; *Majima, 2017*), attentional blink (*Barnhoorn et al., 2015*; *Brown et al., 2014*), flanker (*Simcox & Fiez, 2014*; *Majima, 2017*; *Zwaan et al., 2018*), Simon (*Majima, 2017*; *Zwaan et al., 2018*), lexical decision (*Simcox & Fiez, 2014*), category learning (*Crump, McDonnell & Gureckis, 2013*), memory (*Brown et al., 2014*; *Zwaan et al., 2018*), priming (*Zwaan et al., 2018*), and decision-making tasks (*Berinsky, Huber & Lenz, 2012*; *Brown et al., 2014*). A previous study using auditory stimuli likewise employed crowdsourcing (*Woods et al., 2017*). A recent study recruited infants aged 5–8 months via crowdsourcing and measured their looking time with webcams (*Tran et al., 2017*). These studies have suggested that the effect size of the performance in such tasks is comparable to that in laboratory experiments; hence, crowdsourcing can be used for diverse online experiments with publishable reliability.

However, conventional studies on sensory perception are completed in the laboratory. Moreover, only authors or their laboratory members, who should be well experienced with psychophysical measurements, often participate in experiments on sensory perception. Only a small number of studies have attempted to run sensory perceptual experiments via crowdsourcing. Previous studies have investigated color (*Lafer-Sousa, Hermann & Conway, 2015*; *Szafir, Stone & Gleicher, 2014*) and randomness (*Yamada, 2015*) on the web but used one-time color-matching, color word selection, forced choices (same or different), or magnitude estimation tasks. A few studies have measured the point of subjective equality, sensitivity, or thresholds using psychophysical methods in studies on color perception (*Ware et al., 2018*), volume perception (*Pechey et al., 2015*), size perception (*Brady & Alvarez, 2011*) scene perception (*Brady, Shafer-Skelton & Alvarez, 2017*), and stimulus visibility (*Bang, Shekhar & Rahnev, 2019*). One reason that experiments on sensory perception are rarely conducted online is the necessity for rigorous control over the experimental environment. Online experiments depend significantly on the
participant's own computing environment, and experimenters cannot control the display settings, visual distance (or visual field), or lighting conditions. Thus far, online experiments seem unsuitable for experimental studies that focus on the visual functions of spatial and temporal resolutions. For example, in examining the issue of the temporal aspect of stimulus presentation, researchers have found that stimuli are systematically presented for 20 ms longer than the programed durations (*de Leeuw & Motz, 2016*; *Reimers & Stewart, 2015*). However, the above concerns might be negligible, and crowdsourcing is possibly suitable for perception studies. In this case, a large sample could be recruited to bring sufficient statistical power. Further, large and diverse samples are beneficial for the examination of individual differences in perception studies.

## Aim of the present study

This study focused on measuring low-level visual perception via online experiments. We examined the contrast threshold in vision via online crowdsourcing and laboratory experiments. Contrast threshold is a non-temporal visual capacity that is highly susceptible to the influence of the display condition during measurement. Its measurement needs strict linearization of the output of the display with gamma correction; however, most displays of home PCs are not linearized. Moreover, the viewing distance should vary across the participants in the online condition; the spatial frequency depends on this distance. We believed that a comparison between web and lab measurements of visual contrast thresholds would provide tangible evidence of what online experiments can and cannot test regarding non-temporal aspects of stimulus presentation. If the non-linearity of monitor displays, differences in the viewing distance, and other possible factors comprise a negligible random effect, then the contrast threshold online and in the laboratory would be similar. Another important issue is boredom in the participants. In online experiments, boredom in participants substantially decreases data quality (*Chandler, Mueller & Paolacci, 2014*); many repetitions are likely to induce boredom. Thus, the present study used two types of iteration for online experiments: the repetition and non-repetition conditions. In the former, participants were presented with each trial 10 times per stimulus condition, whereas in the latter condition, each trial was presented only once. If we could control for measurement errors or individual differences by increasing the sample size, then a single trial for a stimulus condition would suffice to lead to an appropriate conclusion, even in online experiments, without data deterioration. For this reason, the sample size of the participants in the non-repetitive condition was about 10 times that of the repetitive condition.

## METHODS

### Participants

We used G*Power to determine the sample sizes needed for the repetition condition ($\alpha = 0.05$, $1-\beta = 0.80$). In the laboratory condition, we used a moderate effect size ($f = 0.25$) in the calculation of the required sample size. The required and maximum sample size was 24. In the online repetition condition, we used a small effect size ($f = 0.10$) in the calculation of the required sample size, because of the potential for noise in the data from

online experiments. The required sample size was revealed to be 138. Considering potential satisficers (*Chandler, Mueller & Paolacci, 2014*; *Oppenheimer, Meyvis & Davidenko, 2009*), who do not devote an appropriate amount of attentional resources to a task and hence cursorily perform it, 200 people was set as the maximum sample size; participants were recruited through a crowdsourcing service (Yahoo! Crowdsourcing: http://crowdsourcing.yahoo.co.jp/). The required sample size in the online non-repetition condition was at least 10 times that in the laboratory condition (240 people) according to the differences in the number of repetitions. Similarly, in the online repetition condition, we recruited 300 people as the maximum sample size to account for the potential influence of satisficers. The participants in the laboratory conditions undertook several experiments, including the present experiment, for 3 h, and subsequently received 4,000 JPY (the present experiment itself took less than 30 min, although we did not accurately record the duration). The order of these experiments was randomized across the participants. The participants in the online repetition and non-repetition conditions received 50 and 20 T-points (Japanese point service, in which one T-point is worth one JPY)[1], respectively. The participants were not made aware of the purpose of the study. The experiment was conducted according to the principles laid down in the Helsinki Declaration. The protocol was approved by the ethics committees of Waseda University (Approval Number: 2015-033) and Kyushu University (Approval Number: 2016-017). We obtained written informed consent from all of the participants in the laboratory condition. Meanwhile, it was difficult to obtain written informed consent in the online conditions. Thus, according to the protocol (Approval Number: 2016-017), we explained the details of the online experiments in instructions sections, and then asked the participants to take part in the experiments only when they agreed to the instructions. We recruited only PC users to participate in the online experiment.

## Apparatus

In the laboratory condition, stimuli were presented on a 23.5-inch LCD display (FG2421; EIZO, Japan). The resolution of the display was 1,920 × 1,080 pixels, and the refresh rate was 100 Hz. We performed gamma correction for the luminance emitted from the monitor. The presentation of stimuli and the collection of data were computer-controlled (Mac mini, Apple, Cupertino, CA, USA). We used MATLAB with the Psychtoolbox extension (*Brainard, 1997*; *Pelli, 1997*) to generate the stimuli. The observer's visual field was fixed using a chin-and-head rest at a viewing distance of 57 cm from the display. The size information at the visual angle described for the laboratory condition was based on this viewing distance. In the online conditions, the experiment was conducted on a web browser with a JavaScript application (jsPsych; *de Leeuw, 2015*). jsPsych is a useful toolbox for psychological research, employed in several previous studies (*de Leeuw & Motz, 2016*; *Pinet et al., 2017*; *Sasaki, Ihaya & Yamada, 2017*).

## Stimuli and procedure

Stimuli consisted of a fixation circle (diameter of 0.24°) and Gabor patches, the diameter of which was 42 pixels (2° in the laboratory conditions). The SD of a gaussian function was

[1] Discrepancies between the number of recruitments and that of the actual participants were often found when we used Yahoo! Crowdsourcing. We could not determine the exact reason. One possibility was that the four-digit number was shared online (e.g., SNS) and some crowdworkers may have seen it. In this case, they could be illegally admitted as completing the task by Yahoo! Crowdsourcing even if they did not actually complete the task. Moreover, Yahoo! Crowdsourcing allowed crowdworkers to access the recruiting page only once. Yahoo! Crowdsourcing manages the number of those accessing the recruiting page via Yahoo ID. Crowdworkers, who had multiple Yahoo IDs, could access the recruiting page several times. Therefore, after a participant had completed our experiment, received the four-digit number, and taken the reward, they could access the recruiting page with their other IDs again and input the four-digit number without performing the experiment. These ways of hacking might have caused the discrepancy. Setting and generating unique four-digit number for each participant could prevent this discrepancy; this is impossible at the present system. We plan to discuss means for preventing these issues with Yahoo! Crowdsourcing.
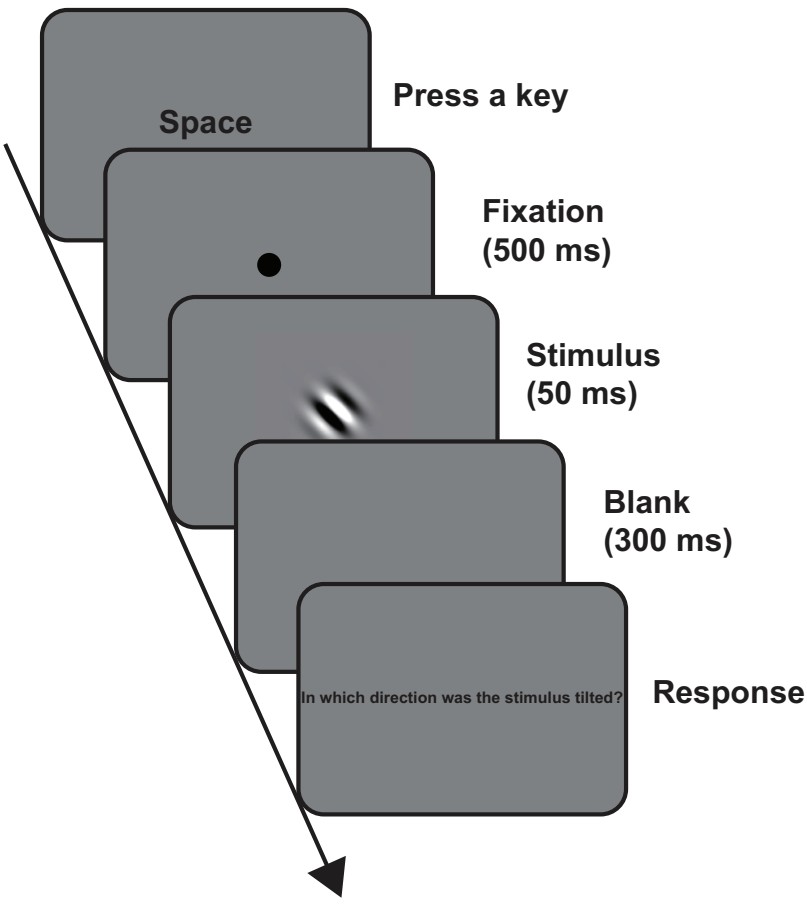

**Figure 1  Timeline of a trial in all the conditions.** For enhanced visibility, we presented the stimulus in 100% contrast level in this figure.               

six pixels (0.29°). There were four spatial frequencies of the carrier: 0.02, 0.05, 0.09 and 0.38 cycles per pixel (cpp; 0.5, 1, 2 and 8 cycles per degree (cpd) in the laboratory conditions). We set seven contrast levels (the Michelson contrast), varying across the spatial frequencies. The contrasts in the 0.02 cpp (0.5 cpd) trials were 3%, 8%, 13%, 18%, 23%, 28% and 33%. The contrasts in the 0.05 and 0.09 cpp (1 and 2 cpd) trials were 1%, 6%, 11%, 16%, 21%, 26% and 31%. The contrasts in the 0.38 cpp (8 cpd) trials were 5%, 10%, 15%, 20%, 25%, 30% and 35%. The Gabor patches were tilted 45° clockwise or counterclockwise. We took screenshots of the stonline non-repetition conditionimuli on the monitor at the laboratory and then used them for the online conditions.

In the laboratory condition, the experiment was conducted in a darkened room. Fig. 1 shows the timeline of a trial in each of the conditions. The participants initiated each trial by pressing the space key. The fixation circle was presented for 500 ms. After the fixation circle disappeared, the Gabor patch was presented for 50 ms. Then, a blank screen was presented for 300 ms, followed by the prompt: "In which direction was the stimulus tilted?" The participants were asked whether the stimulus was tilted clockwise or counterclockwise. They responded without time limits or feedback. Each of the spatial frequency conditions was conducted in a separate session; thus, the experiment consisted

of four sessions. The session order was randomized across the participants. In each session, trials were conducted for seven contrasts in two orientations. In the repetition condition, each combination of contrast and orientation was presented 10 times per session. Thus, participants in the repetition condition completed 560 trials, whereas those in the non-repetition condition completed only 56. The order of the trials was also randomized across the participants. Before the first session, we conducted a practice session, in which the participants completed four trials. The spatial frequency of the practice session was identical to that of the first session, and the contrast was 100%. Both of the orientations appeared twice. As in the experiment conditions, the trial order of each session was randomized across the participants.

In the online conditions, the participants accessed the page of Crowdsourcing for the link to the web address of the experiment. They navigated to the experiment page via the web address and then input their age and sex. Moreover, after completing the experiment, a four-digit number (8,382 and 3,599 in the online repetition and non-repetition conditions, respectively) was presented at the final experiment page; the participants typed this number on an empty form on the Yahoo! Crowdsourcing page. The four-digit number was registered for Yahoo! Crowdsourcing in advance. Only when the input and registered numbers corresponded would Yahoo! Crowdsourcing acknowledge that the participants had completed the experiment and give the reward. If the input and registered numbers did not correspond, Yahoo! Crowdsourcing made the participants drop out and did not give the reward. The procedures were identical to that of the laboratory conditions, except for the added insertion of attention check questions (ACQs). This additional step was included because online participants are often distracted (*Chandler, Mueller & Paolacci, 2014*) or are satisficers (*Oppenheimer, Meyvis & Davidenko, 2009*). ACQs can reduce low-quality responses (*Aust et al., 2013*; *Oppenheimer, Meyvis & Davidenko, 2009*). These tend to be easy calculations based on the four basic arithmetic operations (e.g., 20 + 15 = ?). In the present study, ACQs appeared halfway through the total number of trials in each session and participants selected the correct answer from five options. We conducted the online repetition and online non-repetition conditions from 25 to 28 January 2019 and 29 January to 7 February 2019, respectively.

## Data analysis

We excluded participants who gave incorrect answers to one or more of the ACQs. In the laboratory and online repetition conditions, we calculated the contrast threshold of each spatial frequency for each participant, for which the proportion of "correct" responses was 0.82 (*Cameron, Tai & Carrasco, 2002*; *Lee et al., 2014*), using a probit analysis (i.e., fitting a cumulative Gaussian function to the proportion of "correct" responses as a function of the contrast level). We used the "glm" function in R (3.4.4). The probit analysis provided the means and standard deviations (SDs) of the distributions. Then, we calculated the contrast thresholds using the means, SDs, and the "qnorm" function in R. We excluded participant data when $\beta$ calculated by the probit analysis was a negative value. This negative value indicated a reduction in correct responses as the

contrast level increased. In such cases, we could not calculate the thresholds. We also excluded the data from participants whose contrast thresholds were less than zero or greater than 100% because the contrast threshold should be within this range. In the online non-repetition condition, we used the pooled data from all the participants and then calculated the contrast threshold for each spatial frequency by the same procedure of the repetition condition.

First, to confirm whether the contrast threshold depended on the spatial frequency, we conducted a one-way analysis of variance (ANOVA) on the contrast thresholds, with spatial frequency as a within-participant factor, for the laboratory and online repetition conditions. We set the alpha level at 0.05 and calculated $\eta_p^2$. When the main effects were significant, we conducted multiple comparison tests using Holm's method (*Holm, 1979*). We conducted the *t*-test six times. Therefore, we increased α from 0.008 to 0.05 based on Holm's correction (*Holm, 1979*).

As our purpose was to examine whether the contrast thresholds were different or equivalent between experimental environments in each spatial frequency, we conducted two-tailed Welch's *t*-tests for the contrast thresholds for each spatial frequency. After the *t*-tests, we conducted equivalence tests for the pairs in which the contrast thresholds were not significantly different. For the equivalence tests, we used the TOSTER package in R (*Lakens, Scheel & Isager, 2018*) and set Cohen's *d* to 0.5. We compared the contrast threshold of the laboratory condition and the online repetition and non-repetition conditions; thus, we had to conduct *t*-tests and equivalence test three times at most. Therefore, we set α from 0.017 to 0.05 based on Holm's correction (*Holm, 1979*).

## RESULTS

The results of the proportion of the correct responses and the thresholds in the laboratory and online experiments are shown in Figs. 2 and 3, respectively. We collected data from 24 people in the laboratory condition. In the online repetition condition, of the 200 people recruited, only 80 participated[1]. As this number did not reach the required sample size, we recruited another 200 people and 86 people participated. Hence, we collected data from 166 people in total. For the online non-repetition condition, of the 300 people recruited, only 156 participated. Therefore, we recruited another 250 people and 129 people participated. Hence, we collected data from 285 people in total. We excluded the data from two (one owing to a negative β and the other, for having a contrast threshold greater than 100%), 84 (53 owing to a negative β; 13, a contrast threshold less than 0; 8, a contrast threshold greater than 100%; and 10, wrong answers to ACQ), and 19 (all owing to wrong answers to ACQ) participants in the laboratory, online repetition, and online non-repetition conditions, respectively, based on the rules detailed in the *Data analysis* section. Thus, we submitted the data from 22 (16 males and six females, mean age ± SEM = 21.39 ± 0.39), 82 (54 males, 26 females, and two non-respondents, mean age ± SEM = 43.56 ± 1.04), and 266 (176 males and 90 females, mean age ± SEM = 42.92 ± 0.61) participants in the laboratory, online repetition, and online non-repetition conditions, respectively, for the statistical analyses.

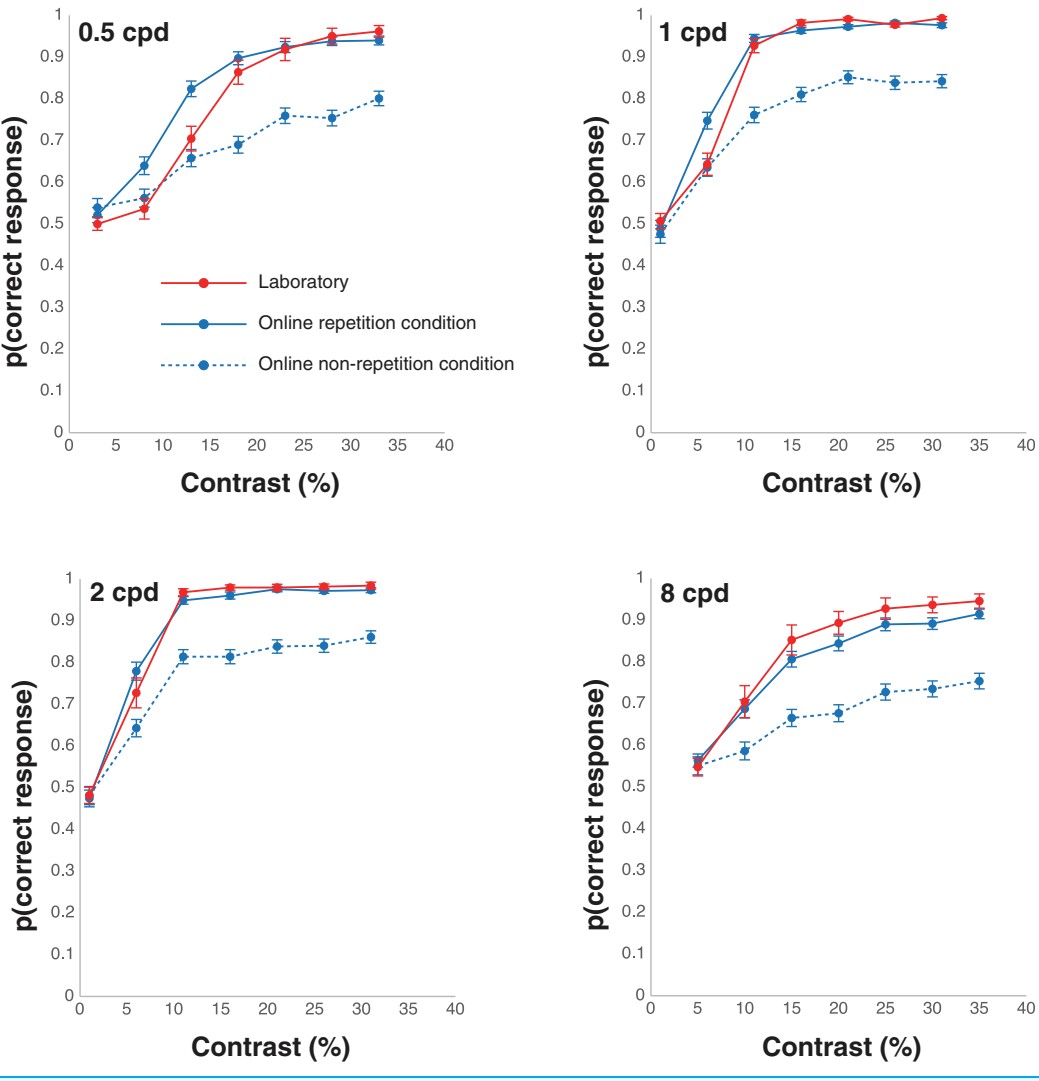

**Figure 2 Results of the correct responses in the laboratory and online experiments.**

## Effects of spatial frequency within the laboratory and online repetition conditions

The results of the ANOVA on the contrast thresholds in the laboratory condition revealed that the main effect was significant, $F(3, 63) = 7.63$, $p < 0.001$, $\eta_p^2 = 0.27$. The multiple comparison tests showed that the threshold was significantly higher in the 0.5 cpd trials compared with the 1 and 2 cpd trials, $ts(21) > 6.25$, $ps < 0.001$, Cohen's $dzs > 1.33$. Moreover, the threshold was significantly higher in the 8 cpd trials compared with the 2 cpd trials, $t(21) = 2.88$, $p = 0.009$, Cohen's $dz = 0.61$. The results of the ANOVA on the contrast thresholds in the online repetition condition revealed that the main effect was significant, $F(3, 243) = 26.23$, $p < 0.001$, $\eta_p^2 = 0.24$. The multiple comparison tests showed that the threshold was significantly higher in the 8 cpd trials compared with the 1 and 2 cpd trials, $ts(81) > 6.77$, $ps < 0.001$, Cohen's $dzs > 0.74$. The threshold was also

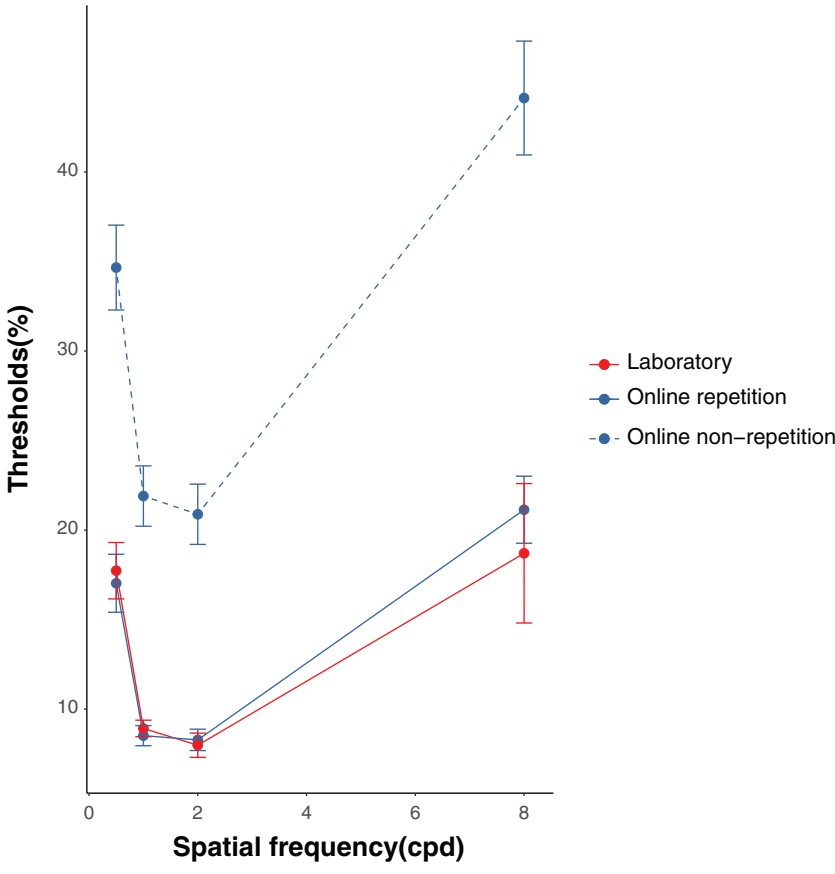

**Figure 3 Results of the thresholds in the laboratory and online experiments.** Error bars denote standard deviations.                                   

significantly higher in the 0.5 cpd trials compared with the 1 and 2 cpd trials, $ts(81) > 4.98$, $ps < 0.001$, Cohen's $dzs > 0.64$. Moreover, we calculated a McFadden's pseudo $R^2$ for each of spatial frequency in the laboratory and online repetition conditions and performed the two-way ANOVA on McFadden's pseudo $R^2$ with spatial frequency as a within-participant factor and experimental circumstances as a between-participant factor[2]. As a result, the main effect of spatial frequency was significant ($F(3, 306) = 27.88$, $p < 0.001$, $\eta_p^2 = 0.21$). Importantly, the main effect of experimental circumstances and interaction were not significant (experimental circumstances: $F(1, 102) = 0.83$, $p = 0.37$, $\eta_p^2 = 0.008$; interaction: $F(3, 306) = 0.52$, $p = 0.67$, $\eta_p^2 = 0.005$).

## Differences and equivalences between laboratory and repeated and non-repeated online conditions

Table 1 shows the summary of the results. For the 0.5 cpd trials, the threshold was significantly higher in the online non-repetition condition compared with the online repetition, $t(332.97) = 6.14$, $p < 0.001$, Cohen's $d = 0.51$, and laboratory conditions, $t(159.41) = 5.95$, $p < 0.001$, Cohen's $d = 0.45$. Meanwhile, the online repetition and laboratory conditions showed no significant difference, $t(68.92) = 0.31$, $p = 0.76$, Cohen's

[2] We added these post-hoc analyses according to the reviewer's comment.

**Table 1  Summary of the results in differences and equivalences between laboratory, repeated, and non-repeated online conditions.**

| cpd | | Laboratory | Online repetition | Online non-repetition |
|---|---|---|---|---|
| 0.5 | Laboratory | | | |
| | Online repetition | **Sig. Eq.**<br>TOST (90% CI [−3.1 to 4.5]) | | |
| | Online non-repetition | **Sig. Dif.**<br>NHST (95% CI [−22.6 to −11.3]) | **Sig. Dif.**<br>NHST (95% CI [−23.3 to −12.0]) | |
| 1 | Laboratory | | | |
| | Online repetition | **Sig. Eq.**<br>TOST (90% CI [−0.8 to 1.6]) | | |
| | Online non-repetition | **Sig. Dif.**<br>NHST (95% CI [−16.4 to −9.5]) | **Sig. Dif.**<br>NHST (95% CI [−16.9 to −9.9]) | |
| 2 | Laboratory | | | |
| | Online repetition | **Sig. Eq.**<br>TOST (90% CI [−1.8 to 1.2]) | | |
| | Online non-repetition | **Sig. Dif.**<br>NHST (95% CI [−16.5 to −9.3]) | **Sig. Dif.**<br>NHST (95% CI [−16.1 to −9.1]) | |
| 8 | Laboratory | | | |
| | Online repetition | **Marg. Sig. Eq.**<br>TOST (90% CI [−9.8 to 4.9]) | | |
| | Online non-repetition | **Sig. Dif.**<br>NHST (95% CI [−35.5 to −15.4]) | **Sig. Dif.**<br>NHST (95% CI [−30.3 to −15.7]) | |

**Note:**
NHST 95% CI = Null Hypothesis Significant Test 95% confidence interval, for cases of a significant difference between pairs; TOST 90% CI = Two One-Sided Test 90% confidence interval, for cases of a (marginally) significant equivalence between pairs.

$d = 0.05$. The equivalence test showed significant equivalence between the online repetition and laboratory conditions, $t(68.92) = 2.26$, $p = 0.013$.

For the 1 cpd trials, the threshold was significantly higher in the online non-repetition condition compared with the online repetition, $t(314.58) = 7.54$, $p < 0.001$, Cohen's $d = 0.55$, and laboratory conditions, $t(285.95) = 7.43$, $p < 0.001$, Cohen's $d = 0.71$. No significant difference was observed between the online repetition and laboratory conditions, $t(82.43) = 0.56$, $p = 0.580$, Cohen's $d = 0.09$. The equivalence test showed significant equivalence between the online repetition and laboratory conditions, $t(82.43) = 2.13$, $p = 0.018$.

For the 2 cpd trials, the threshold was significantly higher in the online non-repetition condition than in the online repetition, $t(319.24) = 7.06$, $p < 0.001$, Cohen's $d = 0.52$, and laboratory conditions, $t(268.92) = 7.11$, $p < 0.001$, Cohen's $d = 0.72$; no significant difference was found between the online repetition and laboratory conditions, $t(57.31) = 0.33$, $p = 0.742$, Cohen's $d = 0.06$. The equivalence test showed significant equivalence between the online repetition and laboratory conditions, $t(57.31) = 2.12$, $p = 0.019$.

For the 8 cpd trials, the threshold was significantly higher in the online non-repetition condition compared with the online repetition, $t(344.97) = 6.23$, $p < 0.001$, Cohen's $d = 0.50$, and laboratory conditions, $t(56.41) = 5.06$, $p < 0.001$, Cohen's $d = 0.51$. However, no significant difference was found between the online repetition and laboratory

conditions, $t(31.40) = 0.56$, $p = 0.577$, Cohen's $d = 0.14$. The equivalence test showed that the equivalence between the online repetition and laboratory conditions was marginally significant, $t(31.40) = 1.48$, $p = 0.075$.

## DISCUSSION

This study examined whether the contrast threshold was properly measured in an online experiment with two conditions: a condition with repetition of trials and another without repetition. The results showed equivalences in the contrast thresholds of the online repetition and laboratory conditions. The contrast threshold in the online non-repetition condition was higher than that in the online repetition and laboratory conditions. Thus, online experiments seem to be able to measure the contrast threshold as adequately as laboratory experiments, provided enough repetition[3]. Notably, it is difficult to measure contrast thresholds without repetitions. However, as discussed below, there was a high rate of exclusions. In this case, it might be difficult to obtain large and diverse data; thus, one of the advantages of crowdsourcing is possibly lost. Taken together, rash decisions to use crowdsourcing for perception studies is likely to be risky at this time.

The present study excluded 51% of the data in the online repetition condition. These exclusions mainly stemmed from the fact that the correct responses decreased as the contrast level increased or the thresholds were under zero. That is, in the online repetition condition, the contrast threshold could be barely calculated precisely. One possibility is that the experimental environment of 49% of the participants in the online repetition condition might be similar to that of the laboratory condition. We were able to calculate the thresholds of these participants and found significant equivalences between the laboratory and online repetition conditions. Meanwhile, the contrast thresholds were much higher in the online non-repetition condition. Although it is difficult to interpret this result, one can argue that the repetitive performance of the experimental task in the online repetition condition caused perceptual learning. It has been well known that contrast discrimination increases with repeated practice or training (*Sowden, Rose & Davies, 2002*; *Yu, Klein & Levi, 2004*). However, there are only 10 repetitions for each stimulus in the online repetitive condition, and this little practice does not seem to cause sufficient perceptual learning. Alternatively, the difference in the results with and without repetition may provide clues for problems specific to online experiments. A large amount of the data was excluded in the online repetition condition. Based on this, we can expect the data obtained via online experiments to be noisy. Such noisy data might be included in and mediate the results of the online non-repetition condition. Given the large amount of data exclusion in the online repetition condition and the results of the online non-repetition condition, we could not conclude that online experiments are adequate for measuring the contrast threshold. Indeed, the contrast threshold would be difficult to measure via crowdsourcing unless the lighting conditions of each online participant can be measured and calibrated via camera.

There may be solutions for improving the situation of online measurements of the contrast threshold. One would be to control the experimental environment of each participant in the online experiments to match that of a laboratory experiment. A previous

[3] In particular, detection to low contrast stimuli on a non-gamma-corrected monitor are often easier than that on a gamma-corrected monitor: There might be differences in performances for the lowest contrast stimuli between laboratory and online repetition conditions. Thus, according to the reviewer's suggestion, we performed a two-way ANOVA on proportions of the correct responses in the lowest contrast stimuli with spatial frequency (0.5, 1, 2 and 8 cpd) as a within-participant factor and experimental circumstances (laboratory and online repetition) as a between-participant factor. As a result, while the main effect of spatial frequency was significant ($F(3, 306) = 3.34$, $p = 0.02$, $\eta_p^2 = 0.03$), the main effect of experimental circumstances and interaction were not significant (experimental circumstances: $F(1, 102) = 0.02$, $p = 0.89$, $\eta_p^2 < 0.001$; interaction: $F(3, 306) = 0.27$, $p = 0.84$, $\eta_p^2 = 0.003$). Thus, at least, the differences in performances for the lowest contrast stimuli were not found in the present study.

study proposed beneficial tips for controlling the size of stimuli, distance from the monitor, sound volume, and brightness (*Woods et al., 2015*). *Woods et al. (2015)* also provided a possible way to adjust color, which seems to be difficult to control across online participants. They referenced the hints from a psychophysical study (*To et al., 2013*) that demonstrated that humans have the ability comparable to a photometer when asked to match two patches in terms of brightness. The potential solution of *Woods et al. (2015)* was to ask participants to video record their computer screen and a colorful object (reference object) close to the screen using the camera on a mobile device, and then manually calibrate the screen color to the reference object. At this time, these methods require much effort from the participants and experimenters, and prone to technological difficulties; thus, they might not be ultimately effective. The ways to control experimental environments easily should lead to a reduction in low-quality data, and to a decrease in the exclusion of data, while also maintaining the ease of online experiments via crowdsourcing.

Another solution is related to participant negligence. In the online experiment, participants might have a difficulty maintaining their motivation while performing tasks; for instance, they may have been unprepared to participate in a psychological experiment and not met the experimenters. Participants with inconsistent motivation often do not devote enough effort to the tasks, and, hence, cursory responses increase (satisficing, *Berinsky, Margolis & Sances, 2016*; *Maniaci & Rogge, 2014*; *Miura & Kobayashi, 2016*; *Oppenheimer, Meyvis & Davidenko, 2009*). ACQs, which we set during the online condition sessions, are beneficial for protecting the quality of the data from satisficing. It is easy for participants to answer ACQs correctly when they perform the tasks carefully. Generally, it is important to exclude the data from those who wrongly answer ACQs because of inattention and/or cursory responses, to improve the quality of data. However, in the present study, the data exclusion owing to incorrect ACQ responses accounted for 6% of the total data in each of the online conditions. Thus, the ACQ might not have worked as intended in the present study. The type of ACQ was extremely different from that of the main task (i.e., judging the orientation of the Gabor patch). Given this, the ACQ could be improved so that participants are not easily caught out, or another method could be used. An instructional manipulation check (IMC) is also helpful for detecting satisficers (*Oppenheimer, Meyvis & Davidenko, 2009*). An IMC checks whether the participants carefully read the instructions for the tasks. Specifically, they can incorporate the instruction not to answer questions into some methods commonly used in psychological research (e.g., Likert scales); thus, if the participants do not carefully read the instructions, they mistakenly answer the questions. The data from such participants should be excluded because they improperly dealt with the tasks. Additionally, in a recent study, alerting satisficers to their inattentiveness by a repeated IMC was helpful in improving their information processing (*Miura & Kobayashi, 2016*). In general, ACQs and IMCs are valid tools for the detection and exclusion of data from satisficers. However, it is difficult to prevent satisficers from participating in experiments. To avoid losing data owing to satisficers, blacklisting them might be more effective in the long term.

Other ways could be employed to maintain the quality of psychophysical online data. One is developing a platform designed for scientific research. Crowdsourcing services, such as Yahoo! Crowdsourcing and Amazon Mechanical Turk, have some advantages for conducting psychological research. However, they were not developed as research tools and have some inconveniences as well. Recently, a platform for scientific research was designed (TurkPrime, recently rebranded as CloudResearch: *Litman, Robinson & Abberbock, 2017*) and integrated with Amazon Mechanical Turk. Prolific is also a remarkable platform for conducting surveys and experiments online (*Palan & Schitter, 2018*). These helpful systems for improving the quality of online data have also been proposed: Excluding participants based on previous participation, communicating with participants, and monitoring dropout and engagement rates. Elevating these platforms should be helpful for improving the quality of data in online experiments.

Contrast sensitivity seemed to be lower in the present study than in the previous ones (*Cameron, Tai & Carrasco, 2002*; *Lee et al., 2014*). This discrepancy might be attributed to the intensity level of the stimulus. Several studies have pointed out that the typical hardware used in psychological studies (256 intensity levels, eight bits) is insufficient for measuring contrast thresholds. One of the solutions is to use a graphics card able to display more than 256 different luminance intensities (*Allard & Faubert, 2008*; *Lu & Dosher, 2013*), but this does not seem to be realistic in online experiments. A previous study proposed the solution of adding visual noise to the stimulus, thereby not requiring special hardware (*Allard & Faubert, 2008*). This solution might fit the context of online experiments. We aim to address these issues in future studies.

Although crowdsourcing does not seem to be suitable for measurements of perception studies at this time, the improvement of environments in online experiments will bring advantages. For example, crowdsourcing enables researchers to obtain large amounts of data from various people, which is advantageous for examining individual differences in perceptual and cognitive processing. In classic laboratory experiments, most participants are university or graduate students, and large amounts of data tend to be difficult to collect. The demographics, personal traits, and cognitive characteristics of the participants do not vary enough to examine the relation between individual differences in perceptual and cognitive processing. Thus, this relation and underlying mechanism have not been understood well, warranting further investigations (*Yamada, 2015*). Crowdsourcing, however, allows researchers to recruit participants from around the world, and hence, mass data from participants with various personality traits can be collected. Indeed, we and others have already shown the relation between individual differences in personality traits (e.g., social anxiety, behavioral activation/inhibition systems, and mood) and emotional reactions using crowdsourcing (*Chaya et al., 2016*; *Sasaki, Ihaya & Yamada, 2017*). Moreover, we previously conducted a perceptual study indicating the age and sex differences in the perception of pattern randomness (*Yamada, 2015*). If the environment in online experiments is improved and crowdsourcing becomes suitable for investigating visual perception, then online experiments will be helpful for addressing issues regarding individual differences in visual perception.

## CONCLUSIONS

The present study examined the suitability of online experiments on the contrast threshold. As a result, online experiments seem to be able to measure the contrast threshold as adequately as laboratory experiments, provided enough repetitions. However, there was a high rate of exclusions, which is likely to spoil one of the advantages of crowdsourcing research. Thus, rash decisions to use crowdsourcing for perception studies might be risky at this time. The improvement of technology environments in online experiments via crowdsourcing will bring advantages; individual differences in perceptual processing will be measurable.

## ACKNOWLEDGEMENTS

We would like to thank Dr. Daiichiro Kuroki for developing the program of the online experiment.

### Funding

This research was supported by JSPS KAKENHI #17J05236 to Kyoshiro Sasaki and #15H05709, #16H03079, #16H01866, #17H00875, #18H04199, and #18K12015 to Yuki Yamada. The funders had no role in study design, data collection and analysis, decision to publish, or preparation of the manuscript.

### Grant Disclosures

The following grant information was disclosed by the authors:
JSPS KAKENHI: #17J05236, #15H05709, #16H03079, #16H01866, #17H00875, #18H04199 and #18K12015.

### Competing Interests

The authors declare that they have no competing interests.

### Author Contributions

- Kyoshiro Sasaki analyzed the data, conceived and designed the experiments, performed the experiments, prepared figures and/or tables, authored or reviewed drafts of the paper, and approved the final draft.
- Yuki Yamada conceived and designed the experiments, authored or reviewed drafts of the paper, and approved the final draft.

### Human Ethics

The following information was supplied relating to ethical approvals (i.e., approving body and any reference numbers):

The protocol was approved by the ethics committees of Waseda University (approval number: 2015-033) and Kyushu University (approval number: 2016-017).

## Data Availability

The dataset is available at Figshare: Sasaki, Kyoshiro; Yamada, Yuki (2019): Online experiment of contrast thresholds. figshare. DOI 10.6084/m9.figshare.9767057.v1.

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
