# Peer review of "Crowdsourcing visual perception experiments: a case of contrast threshold"

_PeerJ, doi:10.7717/peerj.8339_

## Round 0.1 · original submission · Major Revisions

Your manuscript have been revised by 3 independent reviewers and, as you can see, they have several concerns about your study. Particularly, reviewer 2 raised several concerns and the necessity of new experiments to validate your findings. If you decide to revise your manuscript, please, take into account all the points of the reviewers. Note that at I cannot guarantee the acceptance of your manuscript even after revising it.

·

Basic reporting

2.11: Basically - Do not state your opinion
2.13: There are articles discussing the issue of payment. Please refer to published resources.
2.14: 1000 - Based on your abstract, you had a smaller N. Why?
3.20: Impressive list. You may also mention studies that look into auditory stimuli, which relates to your scope. Also, there are more examples of studies made with jsPsych.
4.18: Add subheaders. Here: aim of the study
5.15: No questions on demographics, environment during the experiment (light conditions, input devices) were asked? Why not?
5.21: 138 - Nice. But surprising too, as crowdsourcing allows for much larger N, which is one of the biggest strengths.
6.6: 3 hours - How much time the experiments presented here took? Any control for the randomisation of the order of the experiments within 3h?
7.2: Add an illustration of the stimuli. Describe the timeline of the lab/online experiment
7.17: The method of constant stimuli was used. - More details?
9.10: Rather specific information in this section. Not sure if this is very useful. I'd rather focus on results more.
9.23: Rather small section. I would like to see an analysis from more angles. For example learning effects, a timeline of the experiment with means of results. Correlations?
Please add anonymised results, code used to produce results, and if possible code used to create the study in the supplementary material.
Average time spent on experiment online and in a lab? Are they similar? If not, why not?
10.1: only 80 participated - Why?
10.4: only 156 participated - Why?
10.7: rules detailed - Hard to follow. Please be more specific about who was filtered based on what
10.7: State dates of data collection for the online experiment
12.9: Could you take more measures for more selective recruitment? What about the design of the system in jsPsych? Without more details on the methods, it is hard to judge how universal your results are.
12.18: online non-repetition condition. Although it is difficult to interpret this result, the results of the online repetition condition may inform our understanding. - Not clear
13.11: At this time, these methods seem to require much effort from the participants and thus might not be effective. - Why exactly you did not
13.17: experiments - How do you know that? You did not ask any questions before or after the experiment?
13.18: situations - Not clear. In crowdsourcing experiments in general?
13.19: Satisficing - What?
14.5: another method - You can also filter bases on the time spent to complete.
15.4: Why was this study performed in the end? Are you planning to conduct more research using crowdsourcing? Expand future work.
16.1: Not clear why this needs to go into a footnote

Experimental design

More details on the timeframes of the experiments and illustrations of stimuli needed.

Validity of the findings

Results need to be supplemented by supplementary material. The results section is rather small.

Additional comments

Well-written text with good English. Please look into strengthening the methods and results sections.

Reviewer 2 ·

Basic reporting

no comment

Experimental design

no comment

Validity of the findings

no comment

Additional comments

The contrast detection threshold as a function of the spatial frequency of the stimulus—the contrast sensitivity function (CSF)—is a fundamental measure of visual function. The study tries to assess it in the laboratory and online using crowdsourcing.

To measure CSF in humans, it is known that the intensity levels that can display the typical hardware used in behavioral sciences (256 intensity levels, 8 bits) are not enough (Allard and Faubert 2008; Lu and Dosher 2013). This problem is solved using hardware (Lu and Dosher 2013) or software solutions such as dithering (Allard and Faubert 2008).

It seems that the study did not use (in the laboratory and online) any solution to increase the range of possible intensity levels, which I think indicates that the CSF was not properly assessed. This might explain why the contrast sensitivity found in the study is much lower than the contrast sensitivity found in previous studies (the manuscript lacks literature about contrast sensitivity in humans) and why some participants had 0% contrast sensitivity.

Also, the motivation of the study is not clear to me. The CSF has a characteristic bandpass tuning shape with a peak at 2-6 c/deg. The shape of the CSF could only be the same in the laboratory and online as long as the viewing distance is the same in both situations so the size of the stimuli is matched in the retina.

Allard, Rémy, and Jocelyn Faubert. 2008. “The Noisy-Bit Method for Digital Displays: Converting a 256 Luminance Resolution into a Continuous Resolution.” Behavior Research Methods 40 (3): 735–43.
Lu, Zhong-Lin, and Barbara Dosher. 2013. Visual Psychophysics: From Laboratory to Theory. MIT Press.

·

Basic reporting

This manuscript reported the empirical study that investigated the viability of online experiments with crowdsourcing population in the domain of human sensory perception. Compared to studies of human higher cognition, it remains unclear whether the online experiment is also a powerful tool for studies in of low-level sensory perception, since it requires more precise control of physical environment.
The authors conducted an experiment that compared visual contrast threshold between participants from conventional student sample and online crowdsourcing sample. Results clearly indicated that data from the conventional laboratory experiment and the online experiment with the same number of iterations are equivalent, however data from online without repetition indicated significantly greater contrast threshold than other conditions. Results also exhibited that the data from online experiment may suffer from a high level of data exclusion partly due to lack of experimental control in the presentation of visual stimulus. From these results, the authors (seemed to) conclude that the online crowdsourcing experiment is useful tool for collecting data even in the visual perceptual study, and they also pointed out practical tips on conducting crowdsourced studies in this field.
The empirical study reported here is well-designed and results from the experiment seems to be beneficial for researchers studying human perceptual process. Unfortunately however, the manuscript as it is, failed to make an important contribution in the literature for two reasons. First of all, it seems to me that the central claim of the manuscript failed to show its distinctness. Second, procedures of the experiment (mainly about sampling and data exclusion) and displays of results cannot be understood easily, therefore should be revised accordingly.

Experimental design

Research question is well defined and the aims of the study is original. And the experiment reported here is well-designed and meets ethical standard. However, I would suggest the authors describe the criteria of data exclusion and the number of participants (or trials) excluded due to these criteria in detail.

Validity of the findings

The empirical findings from the present study is straightforward and seems to be statistically sound, although it seems that there’s room for improvement in the description of results for better understanding.
Unfortunately however, conclusion of the manuscript seems not to be explained clearly. I understood that the primary aims of present study is to exhibit the usefulness of crowdsourced online experiments for studies of low-level sensory perception. However, throughout the manuscript, the authors discussed about many cons of crowdsourcing usage. On the other hand, they did not discuss much about the pros of crowdsourcing. In fact, the results may suggest that the crowdsourced experiments on the low-level visual perception cannot be recommended due to its high exclusion rate. However, in the discussion section, the authors tried to conclude that crowdsourcing is a profitable tool for perception studies even if researchers should take into consideration of its high rate of exclusion. If the authors intended to argue benefits, they might want to describe pros of crowdsourcing more in detail throughout the entire manuscript. On the other hand, if they intended to give a warning to rash decisions to use crowdsourcing in low-level perceptual studies, the discussion section should be rewritten accordingly.

Additional comments

Please refer the attached document to see the detailed comments on the above 3 areas and other minor suggestions.

---

## Round 0.2 · Major Revisions

Your manuscript has been revised by two original reviewers and by a new one. The opinion of the reviewers are contradictory and so I may possibly have to send the paper to another round of review. Please, try to answer the original questions of reviewer 2. Otherwise, I will have to contact new reviewers for your manuscript.

·

Basic reporting

The replies of the authors have satisfied my concerns and answered my questions. I think the method of the presented study could be stronger, yet the received results are still worth sharing with the academic world. I support the publication of this article.

Experimental design

Thank you for improving this section and adding the timeline figure.

Validity of the findings

Thank you for improving the results section.

Additional comments

Thank you for considering my comments.

Reviewer 2 ·

Basic reporting

see General comments for the author

Experimental design

see General comments for the author

Validity of the findings

see General comments for the author

Additional comments

My main concern has not been addressed. As I described, the measurement instrument used in the experiments does not have enough sensitivity to measure CSF in humans, which is the main aim of the study.

·

Basic reporting

The revised manuscript has been improved significantly from previous ones. I appreciate authors' extensive works in revising the manuscript.

Experimental design

Research question is well defined and the present study has successfully contributed to the current domain. In the revised manuscript, the criteria of data exclusion and the number of excluded participants has been described in detail. I found that the revision could enhance better understanding of the method.

Validity of the findings

In the revision, the results has been well displayed, and hence improved significantly in its understandability. The conclusions are well stated and compatible with present results. But, I also believe that it will enhance further contribution of this paper if authors discuss about any possible measures to resolve discrepancy issue, in other words, the method to prevent free-riders. For example, it might be possible to generate unique (different, I mean) random completion code to each participant. In this case, researchers should decide who is paid. I recognize that it might cause researchers additional work, but it also seems that it's worth the cost if we consider possible (and quite large number of) free-riders. Furthermore, readers might want to know whether other crowdsourcing providers suffer from similar high rate of free-riding. Any discussion (and suggestion) on this issue might be beneficial to other researchers.

Additional comments

As I mentioned previously, the revision significantly improves the quality of the manuscript. I admire the authors' extensive work on this revision. I thought that my comment on the possible measures to prevent free-riding will be beneficial to research communities, however it is not a requisite condition of acceptance since those measures and their effectiveness is speculative and might be dependent of environment of service provider.

---

## Round 0.3 · Major Revisions

Your manuscript has been evaluated by an additional reviewer and as you can see, the reviewer is asking for more details and new statistical data analysis. Indeed, it is not clear why the authors performed isolated one-way ANOVA. Please, respond the three questions raised by the reviewer and indicate in the revised manuscript where you have introduced the changes indicated by the reviewer. Otherwise, I will need to ask for a re-re-review of your manuscript by the last reviewer.

·

Basic reporting

The authors' response have satisfied my concern regarding free-riders issue. I think that the impossibility of preventing free-riders may restrict usefulness of the current system as a recruitment tool, so I am looking forward to the authors' further contributions on this issue. I support the publication of this manuscript in a current form.

Experimental design

No comments

Validity of the findings

The same comments as 'Basic reporting'

Additional comments

No comments

Reviewer 4 ·

Basic reporting

The manuscript is well written, and the subject is timely. However, it is important not only to show no difference in CSF between laboratory and crowdsourcing conditions but also to address discrepancies in anything other than CSF due to physical differences between the two conditions. Further analyses are required to determine the difference between the two conditions.

Experimental design

no comment

Validity of the findings

1. In general, contrast measurements are significantly affected by gamma correction. In particular, the absence of gamma correction increases the likelihood of detecting stimuli at low contrast.
However, this manuscript did not show the difference in sensitivity between the two conditions (Laboratory vs. Online repetition) at low contrast. Is there a significant difference in mean % correct between the two conditions at the lowest contrast level? It would be appreciated if you performed 2-way mixed model ANOVA (4 cpd x 2 conditions) and showed the results (main effect and interaction).

2. For a better understanding of the results, it would be good to provide figures that show the actual data points at each contrast (i.e., mean accuracy as a function of contrast level). Besides, is there a difference in the goodness of fit (R-square) between the two conditions?

3. The difference between non-repetition and repetition in crowdsourcing conditions can be learning (Fiorentini, A., & Berardi, N. (1980). Perceptual learning specific for orientation and spatial frequency. Nature, 287(5777), 43-44) or testing effect. I hope this possibility is discussed in the text.

---

## Round 0.4 · accepted · Accept

As you can see, the reviewer is recommending the acceptance of your revised manuscript. But they have indicated some minor changes which can be addressed while in production.

Reviewer 4 ·

Basic reporting

I am satisfied that the authors have addressed all my comments and consider that the manuscript is ready for publication.
Minor comment: I recommend adding error bars in figure 2.

Experimental design

no comments.

Validity of the findings

no comments.

Additional comments

no comments.